

# Genome-wide identification and expression analysis of new cytokinin metabolic genes in bread wheat (*Triticum aestivum* L.)

Muhammad Shoaib[1,2,*], Wenlong Yang[1,*], Qiangqiang Shan[1,3], Muhammad Sajjad[1,4] and Aimin Zhang[1]

[1] The State Key Laboratory of Plant Cell and Chromosome Engineering, Institute of Genetics and Developmental Biology, Chinese Academy of Sciences, Beijing, China

[2] University of Chinese Academy of Sciences, Beijing, China

[3] College of Agronomy/The Collaborative Innovation Center for Grain Crops in Henan, Henan Agricultural University, Zhengzhou, China

[4] Department of Environmental Sciences, COMSATS University Islamabad (CUI), Vehari campus, Vehari, Pakistan

[*] These authors contributed equally to this work.

Corresponding authors
Wenlong Yang,
wlyang@genetics.ac.cn
Aimin Zhang,
amzhang@genetics.ac.cn

## ABSTRACT

Cytokinins (CKs) are involved in determining the final grain yield in wheat. Multiple gene families are responsible for the controlled production of CKs in plants, including isopentenyl transferases for *de novo* synthesis, zeatin O-glucosyltransferases for reversible inactivation, β-glucosidases for reactivation, and CK oxidases/dehydrogenases for permanent degradation. Identifying and characterizing the genes of these families is an important step in furthering our understanding of CK metabolism. Using bioinformatics tools, we identified four new *TaIPT*, four new *TaZOG*, and 25 new *TaGLU* genes in common wheat. All of the genes harbored the characteristic conserved domains of their respective gene families. We renamed *TaCKX* genes on the basis of their true orthologs in rice and maize to remove inconsistencies in the nomenclature. Phylogenetic analysis revealed the early divergence of monocots from dicots, and the gene duplication event after speciation was obvious. Abscisic acid-, auxin-, salicylic acid-, sulfur-, drought- and light-responsive *cis*-regulatory elements were common to most of the genes under investigation. Expression profiling of CK metabolic gene families was carried out at the seedlings stage in AA genome donor of common wheat. Exogenous application of phytohormones (6-benzylaminopurine, salicylic acid, indole-3-acetic acid, gibberellic acid, and abscisic acid) for 3 h significantly upregulated the transcript levels of all four gene families, suggesting that plants tend to maintain CK stability. A 6-benzylaminopurine-specific maximum fold-change was observed for *TuCKX1* and *TuCKX3* in root and shoot tissues, respectively; however, the highest expression level was observed in the *TuGLU* gene family, indicating that the reactivation of the dormant CK isoform is the quickest way to counter external stress. The identification of new CK metabolic genes provides the foundation for their in-depth functional characterization and for elucidating their association with grain yield.

## INTRODUCTION

Wheat (*Triticum aestivum* L.) is the predominant cereal crop, second only to rice as the most important staple, with global production nearing 740 million tons of grain (*USDA, 2017*). The projected increase in the human population will increase the production demand to 900 million tons by 2050 (*FAO, 2016*); thus, increasing the yield per unit area will be important for meeting this mounting challenge (*Bartrina et al., 2011*).

Recent studies in model plants have revealed that cytokinin (CK) metabolic genes are strongly associated with plant yield (*Ashikari et al., 2005*; *Bartrina et al., 2011*). CKs are phytohormones that play key roles in regulating the vegetative and reproductive development of plants (*Mok & Mok, 2001*; *Zalewski et al., 2010*). Most CKs are adenine derivatives and have an isoprenoid or aromatic side chain attached to the N-6 of the purine ring (*Avalbaev et al., 2012*). *trans*-Zeatin-, *cis*-zeatin-, and dihydrozeatin-type CKs have also been reported in plants, but their abundance is species-specific (*Sakakibara, 2006*). Genetic manipulation of the genes involved in CK homeostasis can be used for yield improvement, as a significant change in CK content has been observed during grain development in crop plants, including wheat (*Jameson, McWha & Wright, 1982*) and rice (*Ashikari et al., 2005*). CK homeostasis is carried out by several gene families including isopentenyl transferases (IPTs) for biosynthesis, zeatin O-glucosyltransferases (ZOGs) for reversible inactivation, β-glucosidases (GLUs) for reactivation, and cytokinin oxidases/dehydrogenases (CKXs) for degradation (*Song, Jiang & Jameson, 2012*).

IPTs are the gene family responsible for CK synthesis. Two possible pathways of CK synthesis have been proposed: (1) degradation of transfer RNA (tRNA) and (2) *de novo* synthesis. The first pathway is catalyzed by tRNA IPTs (EC 2.5.1.8); However, it is not considered as a major source of CK production (*Takei, Sakakibara & Sugiyama, 2001*). *De novo* biosynthesis of CKs is carried out by adenylate IPTs (EC 2.5.1.27) by adding an isopentenyl group to the N6 terminal domain of ATP (*Frébort et al., 2011*). To date, nine, eight, 11, and six IPT genes have been reported in *Arabidopsis*, rice, maize, and wheat, respectively (*Chang et al., 2015*; *Song, Jiang & Jameson, 2012*). *Kakimoto (2001)* examined the expression of *AtIPT* gene family and demonstrated that *AtIPT3*, *AtIPT5* and *AtIPT7* were expressed in all the tissues, whereas *AtIPT6* and *AtIPT1* were only expressed in the siliques . Moreover, in maize, the tRNA-IPT genes *ZmIPT1* and *ZmIPT10* were reportedly highly expressed in all the organs, whereas the expression patterns of the remaining *ZmIPT* genes were spatially and temporally specific (*Vyroubalová et al., 2009*). In wheat, *TaIPT2*, *TaIPT5*, and *TaIPT8* are expressed during the reproductive stage, and *TaIPT2* exhibits the highest expression level (*Song, Jiang & Jameson, 2012*). Controlled expression of IPT genes can be used to improve plant growth and development (*Faiss et al., 1997*). Transgenic plants harboring a high molecular weight gluten promoter fused with an IPT gene (*HMWipt*) exhibited increased seed weight (*Daskalova et al., 2007*). IPT protein production under the

control of the $P_{SARK}$ promoter in transgenic peanuts led to drought tolerance, delayed senescence and most importantly, a 51–65% increase in seed yield compared with the wild type (*Qin et al., 2011*).

Zeatin, an active form of CK, was first identified in maize. Glycosylation of zeatin to O-glucosyl-zeatin and O-xylosyl-zeatin is carried out by ZOG and O-xylosyl transferase (ZOX), respectively (*Martin, Mok & Mok, 1999*). To date, three ZOG genes in *Arabidopsis*, three in wheat and several ZOG genes in maize have been identified (*Song, Jiang & Jameson, 2012*). O-glucosylation of zeatin-type CKs is reversible in nature. The deglycosylation of zeatin type CKs is catalyzed by GLU (*Brzobohaty et al., 1993*). GLU genes are the members of the glycoside hydrolase 1 family and are involved in the regulation of CK metabolism (*Song, Jiang & Jameson, 2012*). In *Arabidopsis* and rice, 47 (*Miyahara et al., 2011*) and 37 (*Sasaki et al., 2002*) GLUs have been annotated, respectively, whereas in wheat only six *TaGLU* genes have been identified thus far (*Song, Jiang & Jameson, 2012*). The substrate specificity of GLU was found to be conserved in ZOGs (*Falk & Rask, 1995*). As *de novo* synthesis of CK is slow, it is likely that reversible degradation and activation of CKs play important roles in maintaining the total CK pool in plants (*Frébort et al., 2011*).

CKXs (EC: 1.5.99.12) are the only enzymes that permanently degrade CKs by cleaving the N6-unsaturated side chain of the CK to adenine and adenosine in a single step (*Ma et al., 2011*). Two conserved domains involved in the catalytic activity of CKXs have been reported, a FAD binding domain at the N terminus and a CK binding domain at the C terminus of the protein (*Avalbaev et al., 2012*). *Pačes, Werstiuk & Hall (1971)* first reported CKX activity in tobacco, whereas the first CKX gene (*ZmCKX1*) was isolated from maize (*Houba-Hérin et al., 1999*). Since then, many CKX genes have been identified in multiple plant species (*Galuszka et al., 2000*). To date, seven CKX genes from *Arabidopsis*, 11 from rice, 13 from maize and 13 from wheat have been partially or completely identified (*Lu et al., 2015*; *Song, Jiang & Jameson, 2012*). As CKX is a multi-gene family, every member of the family is expected to have specific biochemical properties (*Yeh et al., 2015*), i.e., organ localization, subcellular localization, and substrate specificity. Using gain- or loss-of-function methods, all of the *AtCKX* genes have been functionally studied (*Zalabák et al., 2013*). Detailed expression analysis of *HvCKX* genes has suggested that *HvCKX1*, *HvCKX4*, *HvCKX9* and *HvCKX11* are more highly expressed in developing kernels, and by using RNA interference technology, *HvCKX-1-* and *HvCKX9*-silenced plants were found to produce more spikes and a greater number of seeds (*Zalewski et al., 2014*).

In rice, the production of more CK as a result of reduced *OsCKX2* expression increased the total yield by increasing the number of reproductive organs (*Ashikari et al., 2005*). *Yeh et al. (2015)* used short hairpin RNA-mediated silencing technology to hinder the expression of *OsCKX2* in rice, resulting in an increased number of tillers and increased grain weight. Based on quantitative expression analysis, 12 bread wheat varieties varying in the numbers of grains per spike were found, and the variation was positively correlated with *TaCKX2.1* and *TaCKX2.2* genes (*Zhang et al., 2011*). *TaCKX6a02-D1a*, an allelic isoform of *TaCKX6a02-D1*, was correlated with grain size, grain weight and grain filling rate. These results were also confirmed in 169 recombinant inbred lines (Jing 411× Hongmangchun

21) and 102 wheat varieties under different environmental conditions. A 29-bp insertion-deletion mutation in the 3′ untranslated region was thought to be responsible for this variation. In another experiment, copy number variation in the *TaCKX4* gene linked to *Xwmc1* 69 on chromosome 3AL was associated with grain weight (*Chang et al., 2015*; *Lu et al., 2015*).

In summary, all members of the aforementioned CK metabolic gene families have been identified in model plants, and in-depth functional studies have been carried out. Nevertheless, the gene family members have not yet been completely identified in wheat. The hexaploidy (AABBDD = 42), large genome size ($\sim$17 GB) and complexity of interactions between the three genomes are among the reasons for this lack of information. In this study, we explored new genes belonging to the major CK metabolic families in wheat, laying a foundation for their detailed characterization.

## MATERIALS AND METHODS

### Plant material

*Triticum urartu* seeds treated with 1% $H_2O_2$ were grown in petri dishes. After 5 days, the seedlings were transferred to hydroponic tanks and grown in controlled conditions (25 °C, 16:8 h photoperiod). Half-strength Hoagland solution (*Hoagland & Arnon, 1939*) modified for solution culture was provided, and the nutrient solution was changed twice a week during the course of the experiment. Fifteen days after germination, seedlings were treated with plant hormones: 5 µM 6-benzylaminopurine (6-BA), 0.5 mM salicylic acid (SA), 10 µM indole-3-acetic acid (IAA), 30 µM gibberellic acid (GA$_3$), and 10 µM abscisic acid (ABA) for 3 h, along with the control treatment. A total of 20 seedlings per biological replicate and three biological replicates per treatment were used. Immediately after 3 h treatment, root and shoot tissues were collected and frozen in liquid nitrogen for RNA extraction.

### RNA extraction and cDNA synthesis

Conventional RNA extraction was performed using TRIzol reagent (TIANGEN Biotech Co., Ltd., Beijing, China) (*Chomczynski & Sacchi, 2006*). The purity and quality of the RNA samples were verified using 1% agarose gel electrophoresis. For cDNA synthesis, 1.5 µg of the RNA template was used in a reaction mixture of 20 µL. A FastQuant RT kit (with gDNase) (TIANGEN Biotech Co., Ltd.) was used according to the manufacturer's instructions, with the final incubation time extended to 30 min at 42 °C.

### Isolation of CK metabolic genes

To retrieve new members of the gene families involved in CK metabolism, the homology search approach was used. cDNA sequences and the conserved domains of all previously annotated genes involved in CK metabolism, i.e., IPTs, CKXs, GLUs, and ZOGs from *Arabidopsis*, maize, and rice, were used to query the wheat database (https://urgi.versailles.inra.fr/blast_iwgsc/blast.php). Matched sequences having *E*-values $\leq 2e^{-7}$ were downloaded. A separate preliminary sequence alignment and a phylogenetic tree for each gene family were constructed to clean the duplicate sequences. Using a

BLASTx search of the NCBI database (https://www.ncbi.nlm.nih.gov/), protein structures and conserved motifs specific to each protein family were confirmed. The theoretical isoelectric points (PIs), molecular weights (MWs) (http://web.expasy.org/compute_pi/), and N-glycosylation sites (http://www.cbs.dtu.dk/services/NetNGlyc/) of CK metabolic proteins were also determined.

## Gene structure and phylogenetic analysis

The structures of CK metabolic gene families and the number of introns and exons were determined using the Gene Structure Display Server (http://gsds.cbi.pku.edu.cn) (*Hu et al., 2015*). For phylogenetic analysis, translated amino acid sequences were used, as protein sequences are more conserved among species. Separate ClustalW multiple alignments (*Thompson, Higgins & Gibson, 1994*) of the protein sequences for each gene family were carried out using Bioedit software (*Hall, 1999*). Based on the conserved domains and full-length protein sequences, an unrooted neighbor-joining phylogenetic tree (bootstrap 1,000) was developed using Geneious software (*Kearse et al., 2012*).

## *In silico* promoter analysis

To identify *cis*-regulatory elements in the promoter regions of gene families involved in CK metabolism, 2-kb upstream regions of the translation sites of the respective genes were extracted from the local wheat genomic database. *In silico* promoter analysis was carried out for all the reported genes of the respective multi-gene families. *Cis*-regulatory elements responsive to light, phytohormones, abiotic stress, heat shock and low temperature were considered. MatInspector software (*Cartharius et al., 2005*) based on the PLACE library (http://www.dna.affrc.go.jp/PLACE/) (*Higo et al., 1999*) was used to explore the *cis*-regulatory elements.

## Quantitative expression analysis

As there was significant sequence similarity in the exonic regions of wheat sub-genomes, gene-specific homoeologous quantitative polymerase chain reaction (qPCR) primers for all members of the *TaCKX*, *TaIPT*, and *TaZOG* families were developed. As *TaGLU* is a large family, qPCR primers were designed from selected family members (seven new and seven old genes). The *Ta4045* primer was used as an internal control (*Paolacci et al., 2009*), and SYBR Green I Master Mix (Roche Diagnostics, Indianapolis, IN, USA) was used in the reaction mixture according to the manufacturer's instructions. qPCR was conducted using the LightCycler 480 system (Roche Diagnostics), with an initial denaturation step at 94 °C for 5 min, followed by 45 cycles of denaturation at 94 °C for 10 s, annealing at 58 °C for 10 s, and extension at 72 °C for 20 s. Three biological and two technical replicates were used to reduce the error. Genes with reliably detectable expression are presented here.

## Statistical analysis

The $2^{-\Delta\Delta Ct}$ method was used to calculate the relative expression levels for each treatment (*Livak & Schmittgen, 2001*). Student's *t*-test was used to determine the significant differences in the expression levels between the control and treated samples. All statistical analyses were carried out using Microsoft Excel software.
## RESULTS

### Bioinformatics analysis of CK metabolic genes

Following in-depth mining of the wheat genomic database, 13 *TaCKX*, seven *TaZOG*, nine *TaIPT*, and 32 *TaGLU* genes were identified (Figs. 1A–1D). With few exceptions, most of the identified genes had homoeologues in the A, B, and D sub-genomes; whereas the CKX gene family members *TaCKX12* and *TaCKX13*, the IPT gene family member *TaIPT4*, and the GLU family member *TaGLU21* did not have homoeologues. In wheat, CK metabolic genes were not uniformly distributed among and along the lengths of chromosomes (Fig. 2). Most of the genes resided away from the centromere towards the distal parts and formed CK metabolic gene-rich regions. The maximum number of CK metabolic genes were found on chromosome groups three and two, whereas only one gene (*TaCKX7*) was found on chromosome group six. None of the members of the *TaCKX* or *TaIPT* gene families were found on chromosome group four, nor were any *TaIPT* genes found on chromosome group six. As *TaZOG* is a small gene family, its members resided only on chromosome groups two, three, five and seven. *TaGLU* genes were distributed on all the chromosomes except chromosome group six. Based on wheat reference sequence 1.0, homoeologues of *TaCKX11* and *TaZOG2* were predicted on an unknown chromosome. cDNA sequences of these homoeologues were BLASTed against *Aegilops tauschii* in the Ensemble database and more than 90% homology was found for the respective genes. Based on this, we predicted that these homoeologues belong to the D genome.

Phylogenetic analyses were conducted for each gene family using previously reported *Arabidopsis*, rice, maize, and wheat genes. The results showed that most of the identified genes from wheat had orthologs in other monocot species, that is why CK metabolic genes in wheat were given names according to their homology in related species (Figs. 3A–3D).

We were unable to identify new putative genes in the *TaCKX* gene family, as all of the sequences were identical to previously identified *TaCKX1-11* genes (*Feng et al., 2008*; *Ma et al., 2011*; *Song, Jiang & Jameson, 2012*; *Zhang et al., 2007*). However, we report the full-length *in silico* extraction of the *TaCKX* gene family and rectification of its nomenclature. In the literature, multiple names for a single *TaCKX* gene sequence were observed (Table 1). We have renamed the *TaCKX* gene sequences according to their homology with rice and maize (Fig. 3A). According to this systematic renaming, the *TaCKX* gene family consists of 13 members, with gene structures varying from 0 to 4 introns (Fig. 4A), predicted protein lengths of 516–555 amino acids (aa), PIs of 5.62–8.68, 0–5 glycosylation sites, and MWs ranging from 55.7 to 59.8 kDa (Table 2A). Phylogenetic analysis also revealed a D genome-specific duplication of the *TaCKX2* gene, forming a cluster (Fig. 3A) with more than 85% sequence similarity.

While mining the database for *TaZOG*, *TaIPT* and *TaGLU* gene families, four new *TaZOG* s (*TaZOG1*, *TaZOG2*, *TacisZOG3* and *TacisZOG4*), four new *TaIPT* s (*TaIPT1*, *TaIPT4*, *TaIPT9* and *TaIPT10*) and 25 new *TaGLU* genes were identified (Figs. 1B–1D).

In the *TaZOG* gene family, with the exception of *TaZOG2*, remainder of the *TaZOG* genes consisted of open reading frames (ORFs) with no introns (Fig. 4B). Exploring their predicted proteins revealed lengths of 467–551 aa, PIs of 5–6.93, and MWs of
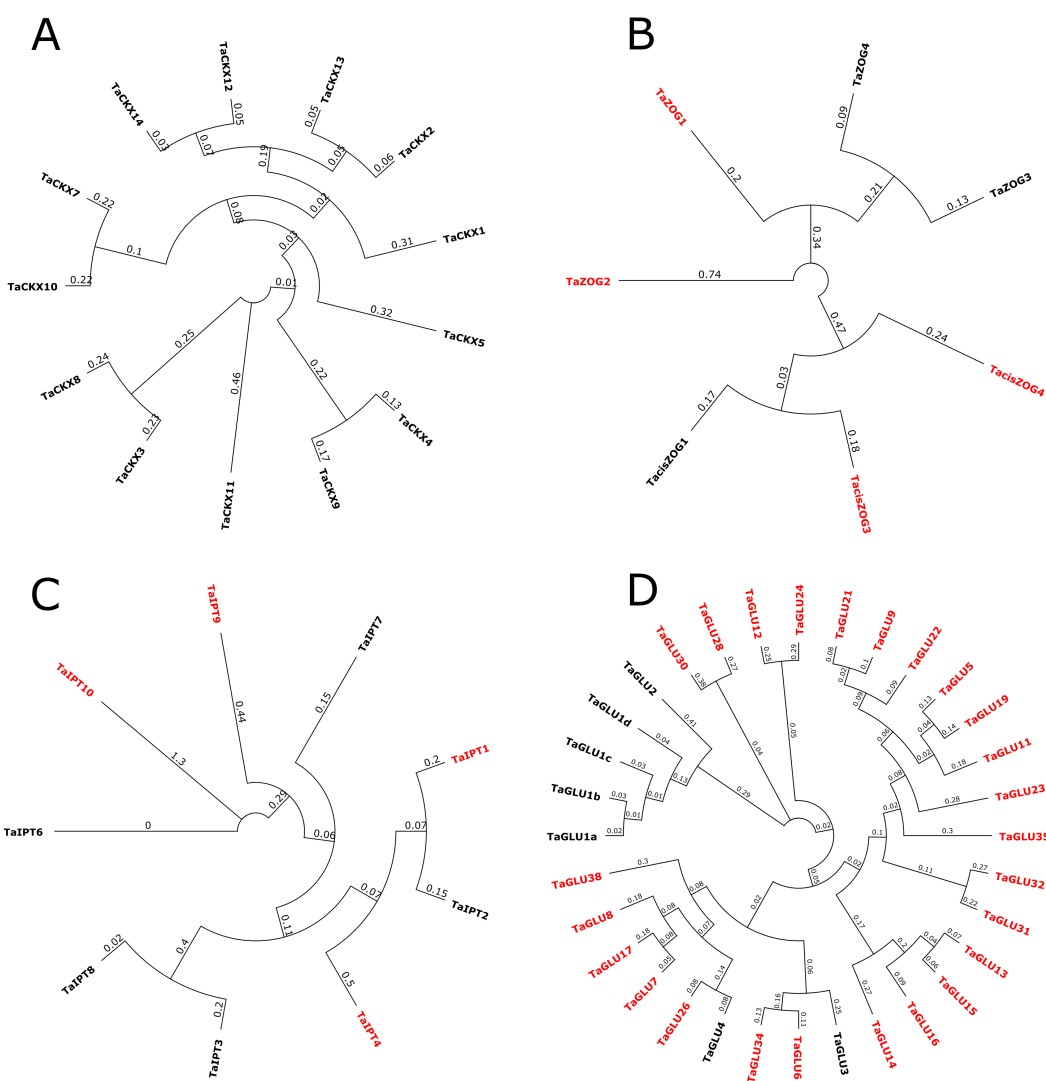

**Figure 1** **The unrooted phylogenetic tree of 13 *TaCKX*. (A), seven *TaZOG* (B), nine *TaIPT* (C) and 32 *TaGLU* (D) genes from wheat.** The tree was produced at amino acid level using neighbor joining method and bootstrapped at 1,000 replications. Newly identified genes are in red color.

50.6–59.3 kDa. Most of the *TaZOG* proteins were predicted to be localized to the plasma membrane, whereas the *cis*-type ZOG proteins were anticipated to be secretory in nature (Table 2B). Genes in the *TaIPT* gene family were also ORFs with no introns, except *TaIPT9* (Fig. 4C). The predicted protein lengths of *TaIPT* genes ranged from 292 to 499 aa, expected MWs from 31.6 to 52.1 kDa, and PIs from 5.05 to 9.24, and the N-glycosylation sites of *TaIPT* s varied from 0 to 1 (Table 2C). Other than the tRNA IPT genes (*TaIPT9* and *TaIPT10*), the rest were predicted to localize to the chloroplast.

In contrast to the IPT family, the GLUs constitute a large gene family. The 25 predicted *TaGLU* genes contained a minimum of 10 introns (Fig. 4D), and the predicted protein size for all *TaGLU* genes varied from 406 to 585 aa. Glycosyl hydrolase family 1/β-glucosidase

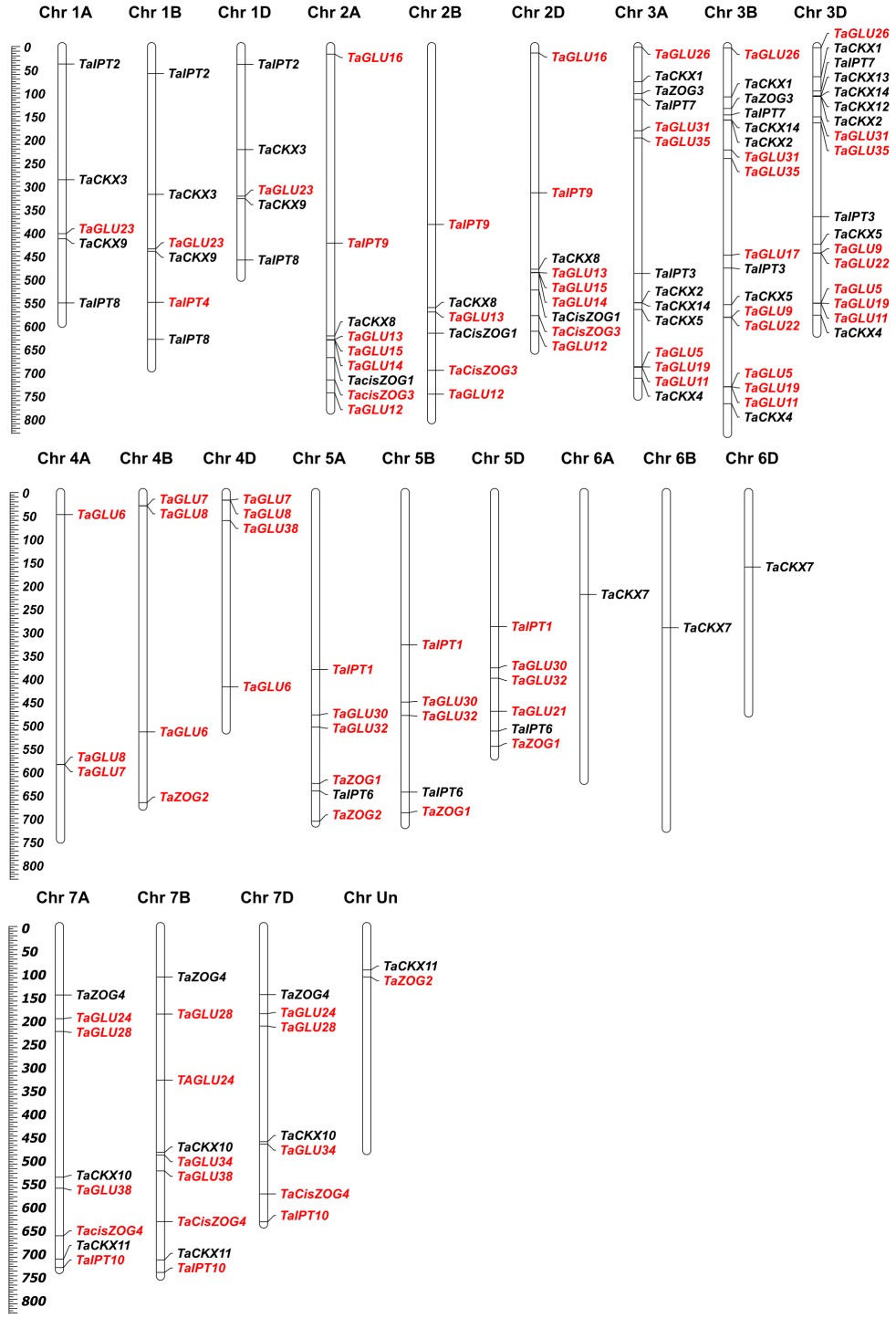

**Figure 2   Chromosome locations of *TaCKX*, *TaZOG*, *TaIPT* and *TaGLU* genes in wheat.** Wheat reference sequence 1.0 was used to develop the physical map of the wheat CK metabolic genes.

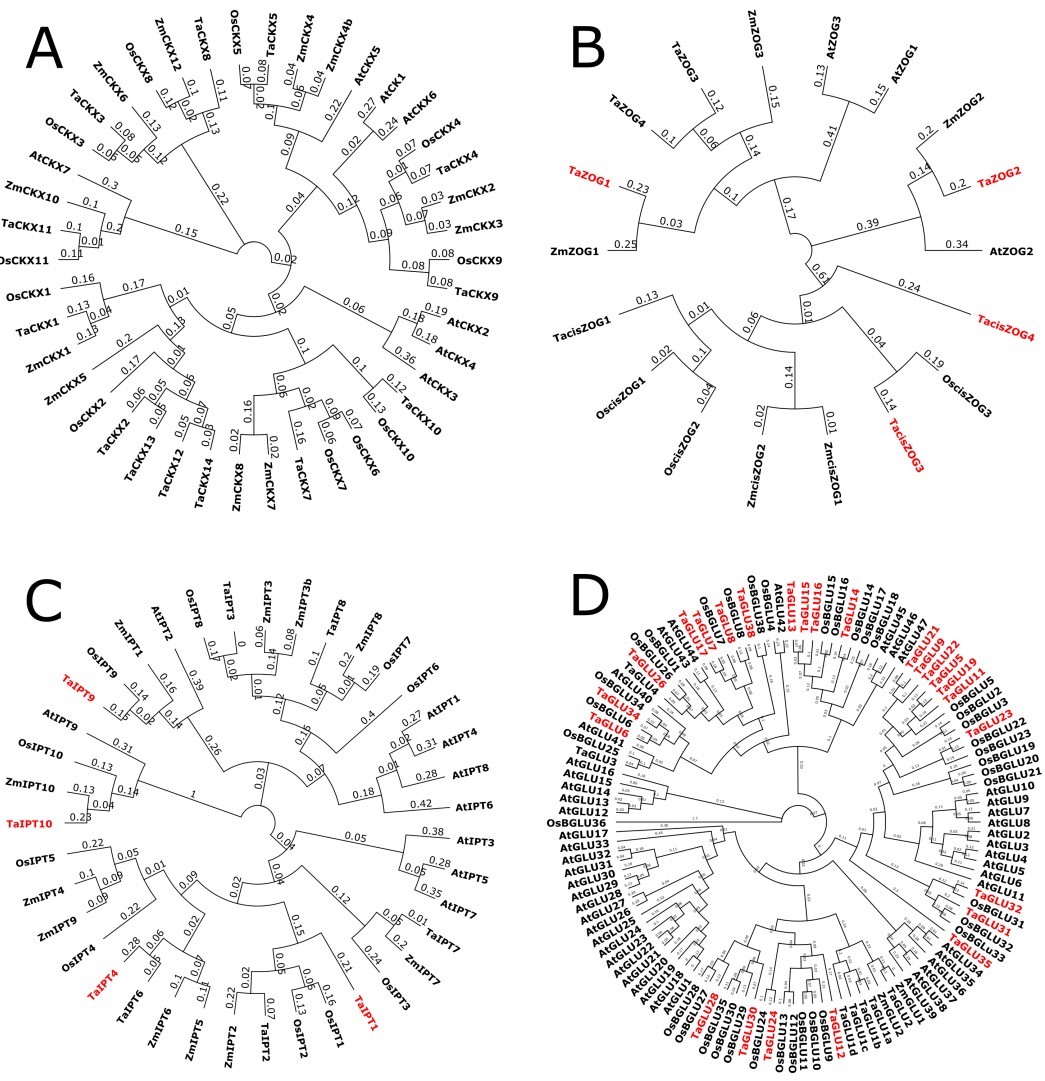

**Figure 3** **The unrooted phylogenetic tree of *CKX* (A), *ZOG* (B), *IPT* (C) and *GLU* (D) genes from *Arabidopsis (At)*, rice (*Os*), maize (*Zm*) and wheat (*Ta*).** The tree was produced at amino acid level using neighbor joining method and bootstrapped at 1,000 replications. Newly identified genes are in red color.

appeared to be the characteristic conserved domain of this family, and the MW was predicted to range from 46.1 to 64.6 kDa. Via *in silico* localization, most of the newly identified *TaGLUs* appeared to be chloroplastic in nature (Table 2D). As wheat is a monocot, newly predicted *TaGLU* genes were homologous to rice rather than the dicot *Arabidopsis*, and this characteristic was clearly mirrored in the phylogenetic analysis of the GLU family (Fig. 3D).

### *In silico* promoter analysis

Promoter analysis of the CK metabolic gene families revealed that drought-responsive *cis*-elements were common to the promoter regions of all members of the *TaIPT*, *TaZOG*,

**Table 1** Previously assigned nomenclature of wheat cytokinin oxidase/dehydrogenase (TaCKX) gene family.

| Sr# | Gene name[a] | Previously assigned nomenclature |
| --- | --- | --- |
| 1 | *TaCKX1* | ***TaCKX1*** (*Feng et al., 2008*; *Song, Jiang & Jameson, 2012*) |
| 2 | *TaCKX2* | ***TaCKX2*** (JN381556.1 GenBank), *TaCKX2.5* (*Mameaux et al., 2012*) |
| 3 | *TaCKX3* | *TaCKX6* (*Song, Jiang & Jameson, 2012*), *TaCKX8* (JQ925405.1 GenBank) |
| 4 | *TaCKX4* | ***TaCKX4*** (*Song, Jiang & Jameson, 2012*) |
| 5 | *TaCKX5* | ***TaCKX5*** (*Lei, Baoshi & Ronghua, 2007*) |
| 6 | *TaCKX7* | *TaCKX8* (*Song, Jiang & Jameson, 2012*) |
| 7 | *TaCKX8* | *TaCKX11* (*Song, Jiang & Jameson, 2012*) |
| 8 | *TaCKX9* | *TaCKX10* (*Song, Jiang & Jameson, 2012*) |
| 9 | *TaCKX10* | *TaCKX9* (*Song, Jiang & Jameson, 2012*) |
| 10 | *TaCKX11* | *TaCKX2* (*Lei, Baoshi & Ronghua, 2008*), *TaCKX3* (*Ma et al., 2010*; *Song, Jiang & Jameson, 2012*) |
| 11 | *TaCKX12* | *TaCKX2.1* (*Zhang et al., 2011*), *TaCKX6D* (*Zhang et al., 2012*) |
| 12 | *TaCKX13* | *TaCKX2.2* (*Zhang et al., 2011*) |
| 13 | *TaCKX14* | *TaCKX2.4* (*Mameaux et al., 2012*) |

**Notes.**
[a] *TaCKX* genes were renamed on the basis of their true orthologs in rice and maize to remove inconsistencies in the nomenclature.
Gene names which matched the new nomenclature are in bold letters.

*TaGLU* and *TaCKX* gene families (Tables 3A–3D). ABA- and sulfur-responsive *cis*-elements were common to members of the *TaCKX* and *TaZOG* families only (Tables 3A and 3B), and cold-responsive *cis*-elements were only found in the promoter regions of *TaGLU* genes (Table 3D).

## Expression analysis

To determine which of the CK biosynthetic and degrading genes were highly expressed or responsive to phytohormones and abiotic stress, the expression patterns of the treated samples were recorded. Genes with reliably detectable expression are presented here. The experiment was conducted on AA genome donor (*Triticum urartu*) of hexaploid wheat. As a basic genome, it has played a central role in wheat evolution and the domestication process (*Ling et al., 2013*).

In general, with the application of phytohormones, the transcript levels of all the genes under study were upregulated as compared to the control treatment, except for *TuGLU4*, *TuGLU9*, and *TuGLU12*, as in roots; their expression levels were drastically lower than that of control treatment (Figs. 5A–5D).

As all the gene families under study are involved in CK metabolism, majority of the genes, with few exceptions, showed significant maximal changes in their transcript levels following exogenous CK treatment. Following CK treatment, most of the genes were responsive to GA₃ treatment, as GA₃ is also a major plant growth regulator; however, their transcript levels varied. While exploring the response of CK metabolic gene families to

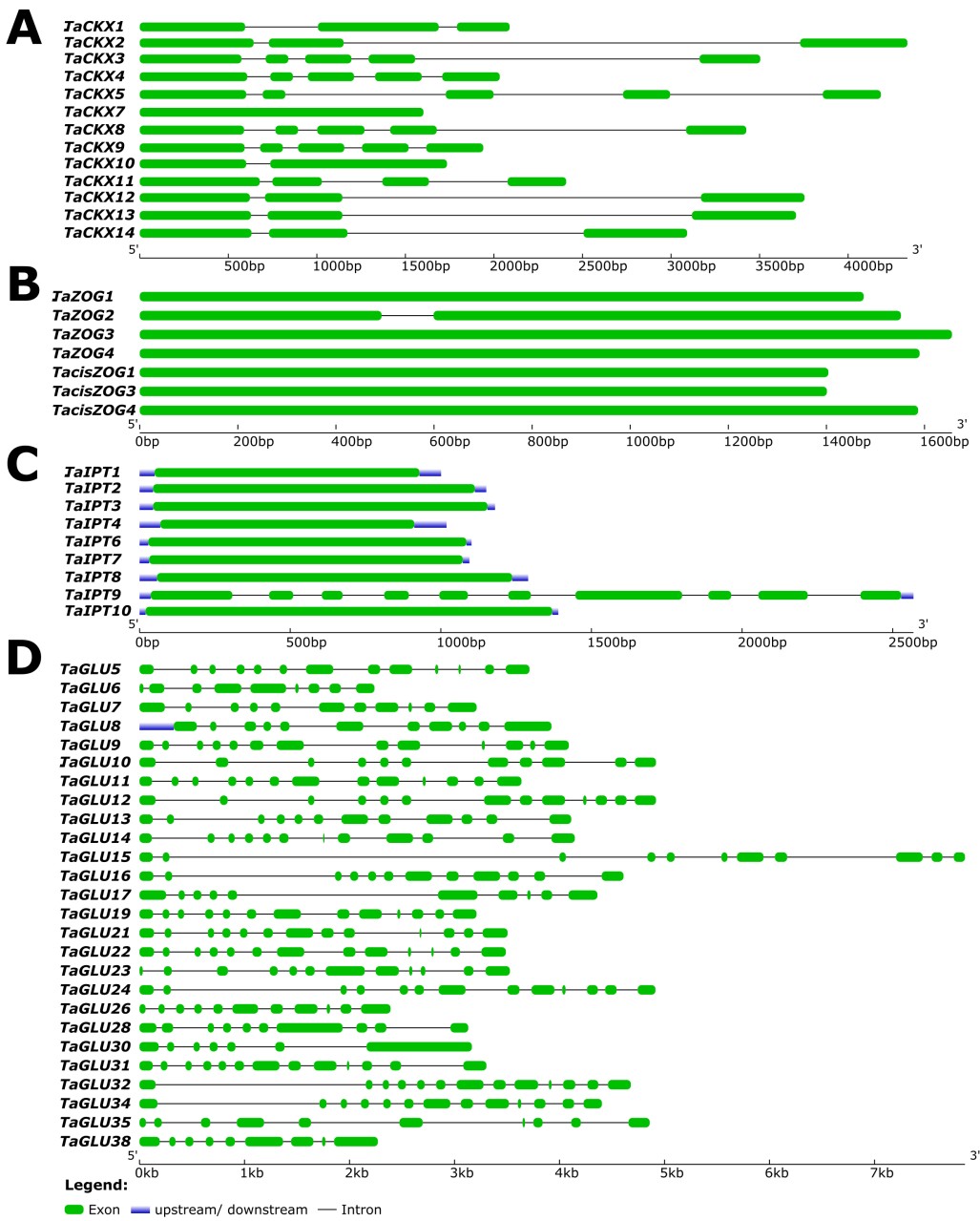

**Figure 4** **Predicted gene structures of wheat *TaCKX* (A), *TaZOG* (B), *TaIPT* (C) gene families and of newly identified *TaGLU* (D) genes.** Exons and introns are illustrated by filled boxes and single lines respectively. UTRs are shown in blue color lines. Gene structures are developed using the Gene Structure Display server (http://gsds.cbi.pku.edu.cn).

ABA treatment, we observed that the mRNA contents of *TuGLU3* and *TuGLU13* in leaf tissues only were significantly higher than those in control plants (Figs. 5B–5D).

For the *TuCKX* gene family, the highest expression level in leaf tissue was recorded for *TuCKX9* (Fig. 5A). In shoots and roots, *TuCKX3* and *TuCKX1*, respectively, showed

**Table 2** Characteristic features of wheat *TaCKX* (a), *TaZOG* (b), *TaIPT* (c) and *TaGLU* (d) gene families.

| | Genes | Length (aa) | PI | MW (kDa) | Subcell location | Glyco. sites |
|---|---|---|---|---|---|---|
| **(a) Cytokinin oxidase/dehydrogenase (*TaCKX*)** | | | | | | |
| 1 | TaCKX1 | 524 | 8.68 | 56.9 | ER & Vacuole | 5 |
| 2 | TaCKX2 | 555 | 6.18 | 59.8 | ER & Vacuole | 2 |
| 3 | TaCKX3 | 523 | 6.28 | 57.7 | ER & Vacuole | 0 |
| 4 | TaCKX4 | 527 | 6.53 | 57.8 | ER & Vacuole | 3 |
| 5 | TaCKX5 | 531 | 6.03 | 57.8 | ER & Vacuole | 2 |
| 6 | TaCKX7 | 535 | 8.49 | 58.5 | ER & Vacuole | 4 |
| 7 | TaCKX8 | 528 | 5.62 | 57.2 | ER & Vacuole | 0 |
| 8 | TaCKX9 | 521 | 6.86 | 58.3 | ER & Vacuole | 5 |
| 9 | TaCKX10 | 532 | 6.1 | 58.0 | ER & Vacuole | 3 |
| 10 | TaCKX11 | 516 | 5.93 | 55.7 | ER & Vacuole | 0 |
| 11 | TaCKX12 | 547 | 5.57 | 59.2 | ER & Vacuole | 1 |
| 12 | TaCKX13 | 545 | 6.05 | 58.8 | ER & Vacuole | 1 |
| 13 | TaCKX14 | 552 | 5.56 | 59.4 | ER & Vacuole | 1 |
| **(b) Zeatin O-glucosyltransferases (*TaZOG*)** | | | | | | |
| 1 | TaZOG1 | 491 | 5.99 | 53.4 | Plasma membrane | 2 |
| 2 | TaZOG2 | 481 | 5.74 | 53.1 | Secreted | 0 |
| 3 | TaZOG3 | 551 | 5 | 59.3 | Plasma membrane | 2 |
| 4 | TaZOG4 | 529 | 5.43 | 56.3 | Plasma membrane | 1 |
| 5 | TacisZOG1 | 467 | 5.87 | 50.8 | Secreted | 0 |
| 6 | TacisZOG3 | 466 | 6.23 | 50.6 | Secreted | 0 |
| 7 | TacisZOG4 | 528 | 6.93 | 57.5 | Secreted | 0 |
| **(c) isopentenyl transferases (*TaIPT*)** | | | | | | |
| 1 | TaIPT1 | 292 | 5.23 | 31.6 | chloroplast | 0 |
| 2 | TaIPT2 | 355 | 5.05 | 38.0 | chloroplast | 0 |
| 3 | TaIPT3 | 369 | 9.24 | 39.2 | chloroplast | 0 |
| 4 | TaIPT5 | 351 | 8.16 | 37.8 | chloroplast | 1 |
| 5 | TaIPT6 | 351 | 8.47 | 37.8 | chloroplast | 1 |
| 6 | TaIPT7 | 346 | 6.68 | 37.1 | chloroplast | 0 |
| 7 | TaIPT8 | 392 | 9.09 | 41.2 | chloroplast | 0 |
| 8 | TaIPT9 | 466 | 6.68 | 52.1 | Cytoplasm | 0 |
| 9 | TaIPT10 | 499 | 6.77 | 50.5 | Cytoplasm | 1 |
| **(d) wheat β-glucosidases (*TaGLU*)** | | | | | | |
| | **Genes** | **Length (aa)** | **PI** | **MW (kDa)** | **subcell location** | **Glyco. sites** |
| 1 | TaGLU5 | 475 | 5.46 | 53.1 | vacuole | 4 |
| 2 | TaGLU6 | 427 | 7.15 | 49 | chloroplast | 2 |
| 3 | TaGLU7 | 508 | 9 | 56.6 | chloroplast | 1 |
| 4 | TaGLU8 | 585 | 6.72 | 64.6 | chloroplast | 2 |
| 5 | TaGLU9 | 532 | 6.9 | 59.5 | vacuole | 2 |
| 6 | TaGLU11 | 508 | 4.91 | 56.6 | vacuole | 3 |

**Table 2** (*continued*)

|  | Genes | Length (aa) | PI | MW (kDa) | Subcell location | Glyco. sites |
|---|---|---|---|---|---|---|
| 7 | *TaGLU12* | 519 | 6.93 | 58.6 | chloroplast | 4 |
| 8 | *TaGLU13* | 508 | 5.7 | 57.5 | chloroplast | 1 |
| 9 | *TaGLU14* | 519 | 6.79 | 59.5 | chloroplast | 2 |
| 10 | *TaGLU15* | 430 | 5.35 | 48.4 | chloroplast | 1 |
| 11 | *TaGLU16* | 511 | 6.12 | 57.8 | vacuole | 1 |
| 12 | *TaGLU17* | 504 | 9.55 | 55.8 | chloroplast | 1 |
| 13 | *TaGLU19* | 506 | 5.36 | 56.7 | vacuole | 2 |
| 14 | *TaGLU21* | 473 | 5.4 | 52.4 | vacuole | 2 |
| 15 | *TaGLU22* | 485 | 5.2 | 53.8 | vacuole | 2 |
| 16 | *TaGLU23* | 477 | 5.6 | 53.4 | vacuole | 2 |
| 17 | *TaGLU24* | 502 | 8.37 | 57.5 | chloroplast | 3 |
| 18 | *TaGLU26* | 448 | 6.67 | 51.5 | chloroplast | 4 |
| 19 | *TaGLU28* | 525 | 8.72 | 59.4 | chloroplast | 4 |
| 20 | *TaGLU30* | 517 | 9.26 | 58 | vacuole | 3 |
| 21 | *TaGLU31* | 503 | 6.05 | 56.5 | vacuole | 6 |
| 22 | *TaGLU32* | 522 | 7.28 | 58.5 | vacuole | 1 |
| 23 | *TaGLU34* | 515 | 6.92 | 58.4 | chloroplast | 4 |
| 24 | *TaGLU35* | 406 | 6 | 46.1 | chloroplast | 2 |
| 25 | *TaGLU38* | 502 | 7.29 | 58.4 | chloroplast | 5 |

**Notes.**

PI, Isoelectric point; MW, Molecular weight; Glyco. Sites, Glycosylation sites; ER, Endoplasmic reticulum.

For *TaGLU* family, characteristic features of only newly identified genes are presented here.

PI & MW predicted by ExPASy (http://web.expasy.org/compute_pi/).

Subcell location predicted by Softberry (http://www.softberry.com/).

Glyco. Sites predicted by NetNGlyc (http://www.cbs.dtu.dk/services/NetNGlyc/).

6BA-specific maximum fold-changes in their expression patterns (Figs. 5A and 5B). In the *TuIPT* gene family, all detectable *TuIPT* genes were upregulated by exogenously applied phytohormones, but the newly identified *TuIPT10* exhibited maximum transcript abundance and was more highly expressed in shoot tissues than in root tissues (Figs. 5A and 5B). In the *TuZOG* family, *TuZOG3* had the highest expression levels and a significant 6BA-specific response in *T. urartu* roots (Fig. 5A).

*TuGLU*s are responsible for the reactivation of reversibly inactivated CKs, and this gene family appeared to be more highly expressed than *TuCKX*s, *TuIPT*s, or *TuZOG*s. Among the *TuGLU*s, *TuGLU7* had the highest transcript level in shoot and root tissues; however, it was significantly responsive to phytohormones only in shoots (Figs. 5B, 5D and Fig. 6B). In contrast, *TuGLU1* appeared to have a root-specific expression and 6BA-specific response (Fig. 5D).

## DISCUSSION

CKs are phytohormones that play important role in the regulation of plant growth. Their role in cell differentiation, nutrient signaling, and leaf senescence have been well established (*Yeh et al., 2015*). Multigene families are reported to maintain CK homeostasis for normal plant growth. In model plants, the genes responsible for CK metabolism have already been identified and well characterized. Using a comparative genomics approach, the conserved

**Table 3** *cis*—regulatory elements in the promoter region of wheat *TaCKX* (a), *TaZOG* (b), *TaIPT* (c) and *TaGLU* (d) gene families.

| | Auxin | SA | ABA | Sulphur | Drought | Cold | Light | GA$_3$ |
|---|---|---|---|---|---|---|---|---|
| **(a) Cytokinin oxidase/dehydrogenase (*TaCKX*)** | | | | | | | | |
| *TaCKX1* | 4 | 4 | 5 | 1 | 9 | 2 | 1 | 1 |
| *TaCKX2* | 0 | 0 | 7 | 4 | 8 | 4 | 0 | 1 |
| *TaCKX3* | 1 | 1 | 1 | 1 | 9 | 2 | 3 | 0 |
| *TaCKX4* | 5 | 5 | 20 | 9 | 22 | 10 | 2 | 1 |
| *TaCKX5* | 1 | 1 | 4 | 4 | 10 | 0 | 2 | 0 |
| *TaCKX7* | 6 | 5 | 7 | 6 | 5 | 6 | 2 | 3 |
| *TaCKX8* | 6 | 6 | 9 | 5 | 6 | 3 | 2 | 1 |
| *TaCKX9* | 0 | 0 | 2 | 3 | 6 | 0 | 4 | 4 |
| *TaCKX10* | 2 | 2 | 7 | 1 | 6 | 2 | 0 | 2 |
| *TaCKX11* | 1 | 1 | 8 | 3 | 8 | 4 | 2 | 0 |
| *TaCKX12* | 5 | 5 | 7 | 4 | 10 | 4 | 0 | 0 |
| *TaCKX13* | 8 | 8 | 12 | 4 | 18 | 9 | 2 | 0 |
| *TaCKX14* | 0 | 0 | 6 | 4 | 11 | 4 | 0 | 1 |
| **(b) Zeatin O-glucosyltransferases (*TaZOG*)** | | | | | | | | |
| *TaZOG1* | 2 | 2 | 10 | 2 | 34 | 5 | 0 | 0 |
| *TaZOG2* | 2 | 2 | 5 | 2 | 9 | 0 | 2 | 0 |
| *TaZOG3* | 0 | 0 | 8 | 6 | 10 | 4 | 2 | 2 |
| *TaZOG4* | 10 | 10 | 15 | 6 | 20 | 6 | 1 | 0 |
| *TacisZOG1* | 7 | 7 | 3 | 3 | 7 | 9 | 0 | 1 |
| *TacisZOG3* | 4 | 4 | 2 | 2 | 6 | 0 | 2 | 0 |
| *TacisZOG4* | 1 | 1 | 4 | 3 | 8 | 2 | 1 | 2 |
| **(c) isopentenyl transferases (*TaIPT*)** | | | | | | | | |
| *TaIPT1* | 2 | 2 | 6 | 2 | 6 | 2 | 1 | 2 |
| *TaIPT2* | 0 | 0 | 1 | 0 | 5 | 0 | 2 | 0 |
| *TaIPT3* | 0 | 0 | 0 | 1 | 3 | 5 | 0 | 1 |
| *TaIPT4* | 2 | 2 | 4 | 2 | 6 | 0 | 0 | 0 |
| *TaIPT6* | 3 | 3 | 2 | 5 | 8 | 6 | 3 | 0 |
| *TaIPT7* | 6 | 6 | 9 | 6 | 12 | 3 | 3 | 1 |
| *TaIPT8* | 2 | 1 | 6 | 3 | 16 | 9 | 2 | 0 |
| *TaIPT9* | 3 | 3 | 5 | 1 | 9 | 4 | 1 | 0 |
| *TaIPT10* | 3 | 3 | 0 | 6 | 8 | 3 | 1 | 1 |
| **(d) wheat β-glucosidases (*TaGLU*)** | | | | | | | | |
| *TaGLU5* | 2 | 2 | 2 | 1 | 6 | 1 | 5 | 1 |
| *TaGLU6* | 4 | 4 | 7 | 4 | 6 | 8 | 1 | 0 |
| *TaGLU7* | 7 | 7 | 9 | 2 | 28 | 7 | 0 | 0 |
| *TaGLU8* | 5 | 5 | 4 | 2 | 8 | 6 | 0 | 0 |
| *TaGLU9* | 6 | 5 | 2 | 4 | 4 | 11 | 0 | 0 |
| *TaGLU11* | 1 | 1 | 3 | 3 | 11 | 3 | 3 | 0 |
| *TaGLU12* | 5 | 4 | 5 | 2 | 5 | 5 | 2 | 0 |

**Table 3** (*continued*)

|  | Auxin | SA | ABA | Sulphur | Drought | Cold | Light | GA$_3$ |
|---|---|---|---|---|---|---|---|---|
| *TaGLU13* | 1 | 1 | 8 | 2 | 22 | 1 | 1 | 0 |
| *TaGLU14* | 4 | 4 | 0 | 3 | 2 | 6 | 1 | 1 |
| *TaGLU15* | 3 | 2 | 5 | 2 | 12 | 2 | 0 | 3 |
| *TaGLU16* | 0 | 0 | 0 | 4 | 9 | 4 | 0 | 0 |
| *TaGLU17* | 5 | 5 | 7 | 1 | 11 | 13 | 0 | 1 |
| *TaGLU19* | 6 | 6 | 8 | 1 | 12 | 3 | 2 | 0 |
| *TaGLU21* | 2 | 2 | 1 | 3 | 5 | 1 | 3 | 2 |
| *TaGLU22* | 2 | 2 | 3 | 4 | 4 | 6 | 1 | 2 |
| *TaGLU23* | 3 | 3 | 20 | 0 | 12 | 16 | 1 | 1 |
| *TaGLU24* | 1 | 0 | 4 | 1 | 6 | 6 | 0 | 1 |
| *TaGLU26* | 2 | 2 | 7 | 1 | 17 | 2 | 2 | 0 |
| *TaGLU28* | 1 | 0 | 5 | 4 | 11 | 3 | 3 | 0 |
| *TaGLU30* | 2 | 2 | 8 | 1 | 11 | 8 | 0 | 0 |
| *TaGLU31* | 1 | 1 | 7 | 3 | 9 | 1 | 5 | 0 |
| *TaGLU32* | 4 | 4 | 7 | 4 | 6 | 4 | 0 | 0 |
| *TaGLU34* | 0 | 0 | 1 | 5 | 5 | 1 | 1 | 0 |
| *TaGLU35* | 2 | 2 | 2 | 2 | 6 | 2 | 1 | 1 |
| *TaGLU38* | 4 | 4 | 6 | 6 | 10 | 3 | 1 | 2 |

**Notes.**

Arabic numerals represent the number of repeats of *cis*- regulatory elements in the promoter region of cytokinin metabolic gene families; whereas, 0 represents absence of specific *cis*-element.

For *TaGLU* family, *cis*- regulatory elements of only newly identified genes are presented here.

PLACE library was used to predict the *cis*-elements, and Auxin, Salicylic acid (SA), Abscisic acid (ABA), Sulphur, Drought, Cold, light and Gibberellic acid (GA$_3$) responsive *cis*-elements were given consideration.

domains and full-length coding sequences of CK anabolic (IPT and GLU) and catabolic (ZOG and CKX) genes from *Arabidopsis*, rice, and maize were used as queries to search the wheat local genomic database. We were unable to identify new genes in the wheat *TaCKX* family. However, for the *TaZOG*, *TaIPT*, and *TaGLU* gene families, we report four, four, and 25 new genes, respectively.

Naming newly identified genes on the basis of their orthologs in closely related species is a systematic way forward (*Lee, Redfern & Orengo, 2007*), as inconsistencies in nomenclature can be misleading (*Goyal et al., 2018*). When reviewing the literature, some irregularities were found in the nomenclature of the *TaCKX* gene family, i.e., multiple naming of homoeologues or single naming of different paralogs (Table 1). The polyploid nature of common wheat and the unavailability of its reference sequence until recently may have led to this discrepancy. In this work, a systematic approach was followed and *TaCKX* gene family members were renamed according to their true orthologs in rice and maize.

Phylogenetic analysis on the basis of sequence similarity is a powerful tool to predict orthologous genes of interest and their functions in important crop species (*Song, Jiang & Jameson, 2012*). *AtIPT2* from *Arabidopsis*, *ZmIPT1* from maize, and *OsIPT9* from rice are actually tRNA IPT genes responsible for the synthesis of zeatin-type CKs in their respective species (*Brugiere et al., 2008*; *Miyawaki et al., 2006*; *Sakamoto et al., 2006*). Based on the sequence and gene structure similarities, newly identified *TaIPT9* in wheat may have a similar function. Phylogenetic analysis also revealed that newly identified *TaGLU* genes

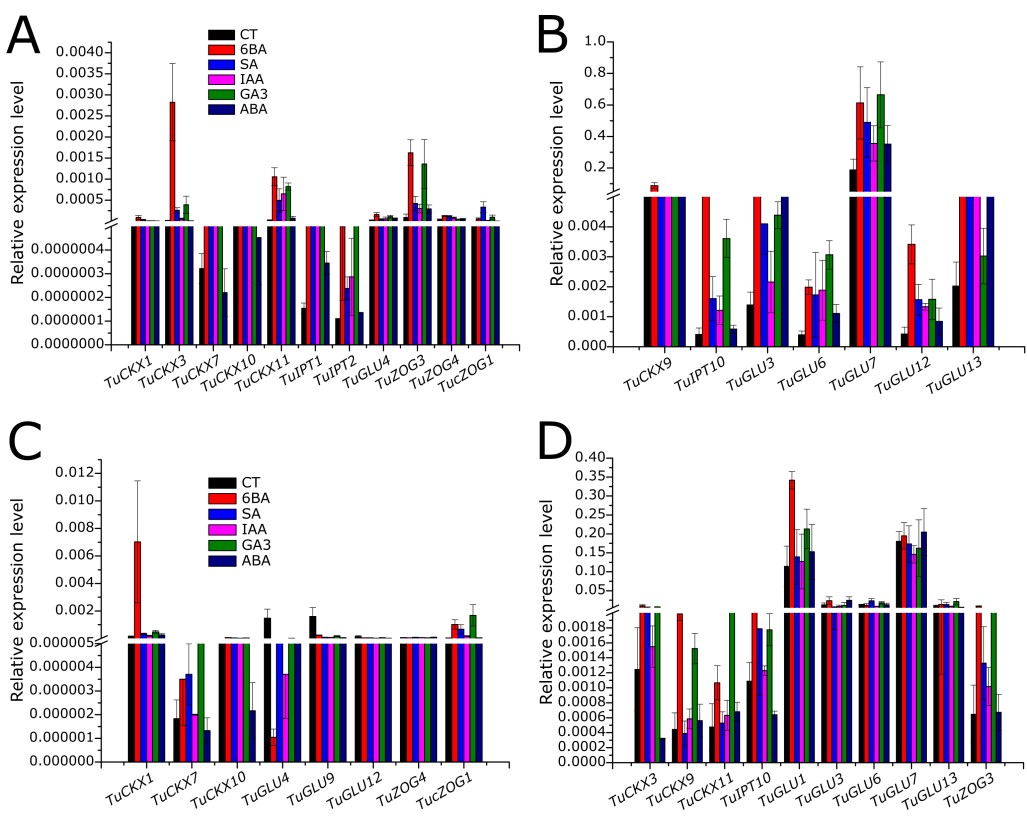

**Figure 5** **Quantitative expression profiles of selected putative cytokinin regulatory genes *TaCKX*, *TaZOG*, *TaIPT*, *TaGLU* in leaf (A) and root (B) tissue of *T. urartu* exposed to exogenously applied phyto-hormones treatment.** (A and C) Selected CK regulatory genes with relatively lower expression. (B and D) Selected CK regulatory genes with relatively higher expression. *Ta4045* gene primer was used as internal control. Two technical and three biological replicates were used to reduce the error. Error bars represent Standard Deviation (*n* = 3).

from wheat are more similar to rice than *Arabidopsis*, depicting the early divergence of monocots from dicot species.

Softberry and NetNGlyc servers were used to predict subcellular localization and glycosylation sites, respectively. Variable subcellular localization and the presence or absence of glycosylation sites within members of each family predicts their variable functions and substrate specificities (*Köllmer et al., 2014*), which will later be confirmed practically. For example, *TaIPT9*, which produces zeatin-type CKs, is predicted to localize to the cytoplasm, in contrast with the remainder of the *TaIPT* genes, which are predicted to localize to chloroplasts.

By controlling the efficiency of gene promoters, *cis*-regulatory elements contribute significantly to the regulation of gene expression. Identifying the targeted *cis*-elements can aid in devising detailed functional studies. Among the putative regulatory elements, ABA-, auxin-, SA-, sulfur-, drought- and light-responsive *cis*-regulatory elements were predicted in most of the promoters of *TaCKX*, *TaIPT*, *TaGLU* and *TaZOG* genes. The broad range

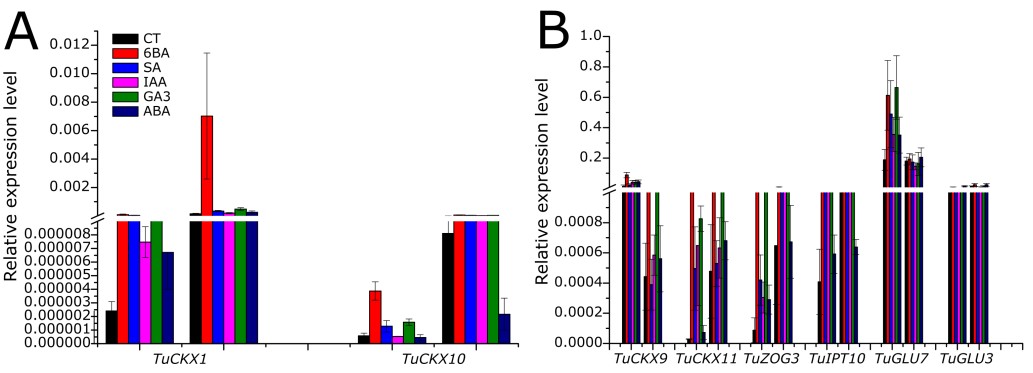

**Figure 6** Comparison among expression profiles of selected cytokinin regulatory genes *TaCKX, TaZOG, TaIPT* and *TaGLU,* in leaf and root tissue of *T. urartu* exposed to exogenously applied phyto-hormones treatment. (A) Selected CK regulatory genes with relatively lower expression. (B) Selected CK regulatory genes with relatively higher expression. *Ta4045* gene primer was used as internal control. Two technical and three biological replicates were used to reduce the error. Error bars represent Standard Deviation ($n = 3$).

of regulatory elements predicts their expression in multiple plant tissues, which may help these gene families stabilize CK content under different environmental stresses.

Before moving forward and carrying out detailed studies of the newly predicted genes, it is necessary to characterize them practically based on expression levels and responsiveness to different stimuli. *T. urartu* seedlings grown under exogenous application of 6-BA, SA, GA$_3$, IAA and ABA hormones were used to develop expression profiles of the above-mentioned gene families. In general, after 3 h of treatment, the transcript levels of all CK metabolic genes were upregulated compared to the control treatment. With the application of external stimuli, CKX genes readily began to degrade active CK. To maintain homeostasis of the CK pool, by feedback mechanism, genes for biosynthetic activity were also triggered. As *de novo* synthesis of CK is relatively slow (*Frébort et al., 2011*), de-glycosylation of O-glycosylated CKs plays a major role in stabilizing CK level (*Vyroubalová et al., 2009*). This can be explained by the higher expression level of *TuGLU* genes compared to those of *TuIPT* genes (Figs. 5A–5D). In contrast to the high expression levels of *TuGLU* genes in leaves and roots under external stimuli, the transcript levels of *TuGLU4*, *TuGLU9*, and *TuGLU12* were antagonistic in both tissues (Figs. 5A–5D). This can be explained by the tissue-specific expression/function of CK metabolic genes (*Vyroubalová et al., 2009*).

In conclusion, we predicted four new *TaZOG*, four new *TaIPT*, and 25 new *TaGLU* genes in wheat and evaluated their sensitivity towards phytohormones. Future studies will be able to mine their biochemical and functional characteristics and their associations with target traits in crop plants.

## ACKNOWLEDGEMENTS

We extend our gratitude to Drs. Linhe Sun, Dongcheng Liu, Jiazhu Sun, Xiaoling Ma and Mr. Dongzhi Wang from Institute of Genetics and Developmental Biology, Chinese

Academy of Sciences and to Dr. Kehui Zhan from College of Agronomy, Henan Agricultural University, for their help with material preparation and manuscript revision.

### Funding

This work was supported financially by the National Basic Research Program of China (2014CB138101) and CAS Strategic Priority Program (XDA08010104). The funders had no role in study design, data collection and analysis, decision to publish, or preparation of the manuscript.

### Grant Disclosures

The following grant information was disclosed by the authors:
National Basic Research Program of China: 2014CB138101.
CAS Strategic Priority Program: XDA08010104.

### Competing Interests

The authors declare there are no competing interests.

### Author Contributions

- Muhammad Shoaib and Wenlong Yang conceived and designed the experiments, performed the experiments, analyzed the data, prepared figures and/or tables, authored or reviewed drafts of the paper, approved the final draft.
- Qiangqiang Shan and Wenlong Yang performed the experiments, contributed reagents/materials/analysis tools, authored or reviewed drafts of the paper, approved the final draft.
- Aimin Zhang conceived and designed the experiments, analyzed the data, authored or reviewed drafts of the paper, approved the final draft.

### Data Availability

The gene and protein sequences of our paper were derived from The IWGSC whole genome assembly CS sequence v1.0 annotation: http://www.wheatgenome.org/.

We also included the data as a Supplemental File.

### Supplemental Information

Supplemental information for this article can be found online at http://dx.doi.org/10.7717/peerj.6300#supplemental-information.

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
