# Peer review of "Genome-wide identification and expression analysis of new cytokinin metabolic genes in bread wheat (Triticum aestivum L.)"

_PeerJ, doi:10.7717/peerj.6300_

## Round 0.1 · original submission · Major Revisions

Besides all comments from the two reviwers, the English language should be significantly improved to ensure your content is clearly and precisely expressed in your text. For example, In several table titles, "cis- regulatory elements responsive to light, ABA, auxins, GA3, SA, drought and low temperature in the promoter region of cytokinin oxidase/dehydrogenase (TaCKX) gene family in wheat". I could not figure out what you try to provide in the Table. Do you want to provide a list of regulatory elements in the table? Are these elements in the promotor region? If so, you should put 'in the promotor region...right after elements. Also, I don't know what are these values in the tables. You need either indicate them in the titles or in foot notes. Are those gene expression levels? Also in several Fig, you did not label what is on the y-axis. Please follow the author's instruction of the journal and make sure style and format meet journal requirement.

Reviewer 1 ·

Basic reporting

no comment

Experimental design

Detailed information for experiment design are needed.

Validity of the findings

no comment

Additional comments

This manuscript reports the identification and characterization of IPT, ZOG, GLU and CKX gene families that are involved in the synthesis and metabolism of cytokinins. Using bioinformatic tools, the authors have predicted 4 new TaZOG, 4 new TaIPT and 25 new TaGLU genes in wheat; and further evaluated their sensitivity towards phyto-hormones. Considering the importance of CKs in regulating vegetative and reproductive development of the plants, the results may provide useful information for the functional characterization of the mentioned gene families in wheat. But there are some points to be considered before publication.

1. Detailed information for experiment design are needed in the M&M section. For example, did the experiment have any replications? Were there any biological replicates of each treatment? How many seedlings were used for each treatment? Were the tissues for each treatment collected at the same time of the day? What is the composition of the Hoagland solution used in the experiment? At least a reference is needed.
2. The English language of the manuscript needs improving. The abstract and discussion sections are well written. But the result section needs to be revised carefully. Too many results were listed in the manuscript, which needs simplification. I encourage the authors to use some summative language instead of simply repeat the information in Tables and Figures. The sentence structure should also be considered for revision. Some detailed information can be found in the attached file.
3. Not much references were referred across the M&M section. Representative references are encouraged to be added for the methods used in the experiments.
4. For references, the full name of journals should be used instead of abbreviations.
5. The manuscript has some careless mistakes. For example,
a) The website supplied for wheat database indicated to an error result. I guess the authors intended to supply this: https://urgi.versailles.inra.fr/blast_iwgsc/blast.php.
b) In the section of “In-silico promoter analysis”, “2000bp” should be “2000 bp”
c) Fig.1, “9 TaIPT genes” instead of “10 TaIPT genes”
Other minor points to be considered can be found in the attached file.

Annotated reviews are not available for download in order to protect the identity of reviewers who chose to remain anonymous.

Reviewer 2 ·

Basic reporting

The English language and typo should be improved for clearly understand the manuscript.

Experimental design

The experimental design requires some proper controls or references.

Validity of the findings

Most of the results do no have proper statistic analysis.

Additional comments

The manuscript describes the identification and expression of cytokinin metabolic genes in wheat. The levels of cytokinins have been shown to regulate grain yield, which can be regulated by IPT, ZOG, GLU and CKX. In this study, the authors identified new IPTs, ZOGs and GLUs genes from bread wheat with bioinformatics tools. Most of these genes are up-regulated the transcript levels in response to different phytohormone treatments. The cis-regulatory elements of these gene families may also link to ABA, auxin, SA, Sulphur, drought and light responses. However, several points are need to be addressed before further process.

1. The readability of the manuscript is needed to significant improvement. For example, Page 9. “Except TaIPT2, TaIPT3 and TaIPT5 cis-regulatory elements related to low temperature were present in all promoter sequences (Table 7).” There is no TaIPT5 in Table 7 and not clear about “all promoter sequences”. In addition, the cold response element also does not present in TaIPT4.
2. In Table 1, based on the results from Song et al., 2012 and Lu et al., 2015, the TaCKX6 gene did not present in this study. Is TaCKX6 same and TaCKX5 or else? Please clarify and consist with the name in all gene families (Table 1-4).
3. In Table 5-8, the number of cis-regulatory elements in the promoter region of cytokinin metabolic gene families may need to compare with the up-stream region of other none related genes. The comparison can be used as a reference for the frequency of each cis-regulatory element.
4. The description of promoters may be more suitable for 2kb upstream region of the translation sites.
5. In Figure 5-7, the relative expression levels of each gene are very different. It would be necessary to describe the amount of total RNA input or the reference gene Ta4045- Otherwise it would be difficult to compare between different genes. Please also add the statistical analysis to the results.
6. Please address the issues why only analyzed portion of the cytokinin metabolic genes (Figure 5, 6, 7). Some the gene expression in response to the phytohormones had been studied prevouly. It would be more meaningful to address the expression of those newly discovered genes or the whole gene family.
7. Thank you for providing the raw data of qRT-PCR, however the calculation of expression ratio is hard to understand. Why the equations for relative expression level needs to *1000 and the standard deviation needs to *LN(2) ? Some of the samples only present two repeats may not sufficient for statistic assay.
8. In abstract, the following statement may need to reconsideration- “Maximum fold change was observed for TaGLU gene family conferring that reactivation of dormant CK isoform is the quickest way to counter the external stress.” The relative expression level of TaGLU7 in the leaf and TaGLU14 in the root after BA treatment are the highest relative to other genes. However, the expression levels only 4-fold and 3-fold different respectively from the control-treatment.
9. Please add the detail statistical analysis including sample number and significant test.
10. Please provide more detail description for each figure/ table legend.
11. The quality of figures (Figure 5-7) is needed to improve, including the data arrangement.
12. It is highly recommended that the number 9 and numbers under 10 should be spelled out.
13. All the gene name should be in italics.
14. The English language and typo should be improved for clearly understand the manuscript. Page 6, Line 5- the RNA quality was check using “1% agarose gel electroporation”- should be “gel electrophoresis”. Page 9, Line 12- “Conversely, cis-elements related to gibberellic acid were confined to TaZOG4; and low temperature responsive elements were bound to TacisZOG1 only.”- not sure the meaning of “confine” and “bind” in the description.

---

## Round 0.2 · Minor Revisions

The revised manuscript significantly improved readability. However as suggested by reviewer 1 that Figs 5-7 have several issues. 1. the labels do not match with Figures. Each Figure has 4 sub-Figures (A-D), but only two can be found (A,B) ; 2.label was wrong, for example, in Figure 5, TaCXX is in sub-Figure A, but they were in. both A and B. Please check for consistency among Figure caption, manuscript and Figure labels. Figures 6 and 5 can be deleted and keep Figure 7. Fig 7 title changes to "Comparison among expression profiles of selected cytokinin regulatory genes TaCKX (A), TaZOG (B), TaIPT (C), TaGLU (D) in leaf and root tissues of T. urartu exposed to exogenous phyto-hormones". Also combine tables 2-5 and Table 6-9 as two tables by adding sub-title in each table for each gene family as suggested by reviewer 1.

Reviewer 1 ·

Basic reporting

no comment

Experimental design

no comment

Validity of the findings

no comment

Additional comments

The manuscript has been well improved. I have gone through the revised manuscript and find that most of the suggestions, comments and corrections by the reviewers have been incorporated. I suggest the manuscript be accepted for publication after the consideration of the suggested issues.

Line 140, “Immediately after treatment” contradicts with the “3 h treatment” mentioned above.

The captions of Figs 5, 6 and 7 indicated four (A B C D) parts for each figure. Whereas, only A and B parts are found in the figures. Furthermore, it may look better if the figures 5, 6 and 7 are combined together, so do Tables 2-5 and Tables 6-9.

Line 567 and Page 45, “7 TaZOG (B), 9 TaIPT (C),” should be “seven TaZOG (B), nine TaIPT (C),”.

Some references in the reference list are not referred in the manuscript, such as Miyawaki et al. 2004, Tsai et al. 2012, and Zalewski et al., 2003. The authors should check the manuscript again to make sure the correspondence.

Reviewer 2 ·

Basic reporting

Please be consistent with the terminology throughout the manuscript.
Table 6-8, line138, 278, 279, 337: "GA3" or "GA[subscript]3".
Figure 5-7, line 345-349, The abbreviation of Triticum urartu is "Ta" or "Tu"?

The image quality of Figure 4 is too low.

Experimental design

No comment

Validity of the findings

No comment

Additional comments

The authors had significantly improved the manuscript from the previous version.
A few minor points are still needed to fix before the following processes.

---

## Round 0.3 · accepted · Accept

Please let me know if you have any question on the review process of this submission.

#